# A Two-Step Approach for 3D-Guided Patient-Specific Corrective Limb Osteotomies

**DOI:** 10.3390/jpm12091458

**Published:** 2022-09-06

**Authors:** Nick Assink, Anne M. L. Meesters, Kaj ten Duis, Jorrit S. Harbers, Frank F. A. IJpma, Hugo C. van der Veen, Job N. Doornberg, Peter A. J. Pijpker, Joep Kraeima

**Affiliations:** 13D Lab, University Medical Centre Groningen, University of Groningen, 9713 GZ Groningen, The Netherlands; 2Department of Surgery, University Medical Centre Groningen, University of Groningen, 9713 GZ Groningen, The Netherlands; 3Department of Orthopaedic Surgery, University Medical Centre Groningen, University of Groningen, 9713 GZ Groningen, The Netherlands

**Keywords:** virtual surgical planning, 3D printing, 3D technology, three-dimensional, corrective osteotomy, osteotomies, malunion, patient-specific, patient-specific instruments, surgical guide

## Abstract

*Background:* Corrective osteotomy surgery for long bone anomalies can be very challenging since deformation of the bone is often present in three dimensions. We developed a two-step approach for 3D-planned corrective osteotomies which consists of a cutting and reposition guide in combination with a conventional osteosynthesis plate. This study aimed to assess accuracy of the achieved corrections using this two-step technique. *Methods:* All patients (≥12 years) treated for post-traumatic malunion with a two-step 3D-planned corrective osteotomy within our center in 2021 were prospectively included. Three-dimensional virtual models of the planned outcome and the clinically achieved outcome were obtained and aligned. Postoperative evaluation of the accuracy of performed corrections was assessed by measuring the preoperative and postoperative alignment error in terms of angulation, rotation and translation. *Results:* A total of 10 patients were included. All corrective osteotomies were performed according to the predetermined surgical plan without any complications. The preoperative deformities ranged from 7.1 to 27.5° in terms of angulation and 5.3 to 26.1° in terms of rotation. The achieved alignment deviated on average 2.1 ± 1.0 and 3.4 ± 1.6 degrees from the planning for the angulation and rotation, respectively. *Conclusions:* A two-step approach for 3D-guided patient-specific corrective limb osteotomies is reliable, feasible and accurate.

## 1. Introduction

Corrective osteotomy surgery for long bone anomalies can be very challenging since the deformation of the bone is often present in three dimensions. Conventional planning methods use two-dimensional (2D) imaging to plan the osteotomy and subsequent surgery is performed freehand, which leads to unpredictable results. With rapid advances in three-dimensional (3D) printing technologies, surgeons have started to apply 3D printing for a wide range of applications in orthopedic trauma surgery [1]. Particularly in corrective osteotomy surgery, the use of 3D-printed surgical guides is well-described and shows promising results in terms of functional outcome and reduced operating time [1,2,3,4]. The use of 3D virtual surgical planning allows the surgeon to visualize the anatomy in 3D, and virtually plan the osteotomy based on the CT scan. Additionally, patient-specific instruments can be designed and 3D printed to guide the cutting and reduction process during surgery. This process takes into account the specific anatomy of the patient and the desired surgical approach, which might lead to a more accurate result [5].

In recent years, different respective methods and techniques for corrective osteotomies of various mechanisms of deformities (i.e., post-traumatic deformities, growth disturbances, congenital anomalies) have been described. The majority of these methods consist of a surgical guide with a cutting slot for the planned osteotomy plane and drilling holes for preplanned screws [6,7,8,9,10,11,12]. Since the screw holes are predrilled, this technique requires an adequate fit of the preplanned plate in order to achieve the planned correction. However, due to the deformity of the bone, conventional plates usually do not fit, which potentially compromises accuracy and can lead to impaired functional outcome. For some cases, precontouring the plate provides a solution [11]. However, bending of a plate does not always result in a good fit; therefore, another solution for adequate use of this 3D technology may be provided by using a patient-specific plate [12]. Yet, currently, patient-specific plates are not widely available; they may be costly and pose logistical and legal challenges. An alternative reported approach is the use of a surgical cutting guide in combination with a reduction guide. The correction is then controlled by placing Kirschner wires (K-wires) with the cutting guide and subsequently realigning of the K-wires towards a parallel position with the use of a reduction guide. However, the combined use of both a reduction guide and a plate is often limited by the small surgical working space. This technique is only described for a few applications [13,14,15,16].

We present this alternative strategy, which consists of our two-step approach for 3D-planned corrective osteotomy, which has been successfully clinically applied. Our method adds to previous reports because a reduction guide, which envelops the planned osteosynthesis plate, is introduced, and therefore the technique is less limited by the surgical working space. This method can be applied disregarding the deformation or location, and is based on our experience in 3D-planned corrective osteotomy surgery over the past few years. This study aimed to assess the accuracy of the achieved corrections using this technique.

## 2. Materials and Methods

### 2.1. Patients

All patients (≥12 years) treated for post-traumatic malunion with a single- or double-cut 3D-planned corrective osteotomy within our center between January and December 2021 were prospectively included upon availability of a pre- and postoperative CT scan with a slice thickness of less than 1 mm. The institutional review board of our center approved the study procedures, and the research was performed in accordance with the relevant guidelines and regulations (registry: 202100639). Written consent was obtained from all patients.

### 2.2. 3D Virtual Surgical Planning of the Corrective Osteotomy

For all patients, a CT scan of the malunited as well as the corresponding contralateral uninjured bone was available. The DICOM (Digital Imaging and Communications in Medicine) image data were imported into the Mimics Medical software package (Version 21.0, Materialise, Leuven, Belgium) in order to create a 3D reconstruction of the affected bone and its counterpart. A segmentation process was performed using a preset bone threshold (Hounsfield unit ≥ 226) combined with the ‘region growing’ and ‘split mask’ function in order to separate the bone from adjacent bones. After the segmentation process, the 3D models of both the malunited and the contralateral bone were imported into the 3-matic software (Version 15.0, Materialise, Leuven, Belgium). The contralateral bone was then mirrored and aligned on an unaffected part of the malunited bone in order to measure the deviation. Based on the deviation, the osteotomy and correction were planned, and a virtual model of the osteosynthesis plate chosen by the surgeon was imported and positioned on the corrected bone. K-wires, at least two on each side of the osteotomy plane, were then placed parallel on the bone after virtual correction, duplicated and reversed engineered towards their original position on the ‘uncorrected’ malunited bone. A cutting guide was then designed, which included the planned osteotomy and the position of the K-wires before correction. In addition, a reposition guide was designed to be placed on top of the planned plate, fitting the K-wires as positioned after the planned correction (Figure 1). During this workflow, multiple interdisciplinary meetings between technical physicians and (orthopedic) trauma surgeons were held to determine surgical approach, level of osteotomy and desired plate positioning in order to ultimately meet our patient’s clinical needs.

### 2.3. Surgical Procedure

After designing the guides, the patient-specific cutting and reposition guides were 3D-printed by selective laser sintering using polyamide 12 (PA12). Additionally, real-size models of the malunion and planned correction were printed 1:1. These prints were sterilized and used during surgery. Exposure of the affected bone was obtained during surgery using a surgical approach as discussed with the surgeon during the stepwise 3D planning process. The cutting guide was then fitted on the bone using bony landmarks, and the K-wires were then placed through the guide (Figure 2c and Figure 3c). The unique footprint directs the guide to the intended location. Positioning of the guide was confirmed by verifying the position of the K-wires with respect to the bony landmarks using fluoroscopy. In the case of incorrect positioning, the guide was repositioned until the surgeon was confident about the correct positioning after repeated visual inspection and radiographic confirmation. The osteotomy was performed through the cutting slot of the cutting guide using an oscillating saw. Subsequently, the planned correction was performed by aligning the K-wires to a parallel position. This process was controlled by sliding the reposition guide, which enveloped the plate, over the K-wires (Figure 2d and Figure 3d). In opening-wedge high tibial osteotomy cases, the planned wedge was incorporated into the guide design for additional strength of the construct and to prevent K-wires from bending during the opening of the wedge (Figure 3). After correction, the design of the reposition guide allowed for at least two screws to be drilled and placed both distal and proximal to the osteotomy level. After placement of these screws, the reposition guide was removed, and the remaining screws were placed. The reposition guide was designed in such a way that the construct of the guide with the fixed K-wires did not block the drilling and placement of the screws.

### 2.4. Postoperative Evaluation

Postoperative evaluation of the accuracy of the performed correction was performed by superimposition of the plan and the postoperative outcome. Subsequently, the preoperative and postoperative alignment error were measured in terms of angulation, rotation and translation. A 3D model of the bone before correction was retrieved from the initial planning. This model was duplicated and aligned with the planned outcome and the postoperative 3D model, such that there were three identical parts with different alignments (preoperative, planned and postoperative parts). In order to measure the angulation and rotation, the inertia axes were automatically drawn using the ‘create analytical primitive’ function in the 3-matic software (Figure 4a). The angulation was then measured as the difference in angle in the z-axis (Figure 4b) and the rotation as the difference in angle in the x-axis (Figure 4c). The translation was obtained by measuring the Euclidean distance in millimeters between the center of gravity of these three parts (preoperative, planned and postoperative). In addition, the clinical outcome was assessed by evaluating the range of motion before surgery and six months after surgery. A Mann–Whitney U test was performed to assess the difference between planned and obtained angulation, rotation and translation. A *p*-value of <0.05 was considered statistically significant.

## 3. Results

### 3.1. Patients

A total of 10 patients, treated for their post-traumatic malunion with a 3D-planned corrective osteotomy, were prospectively included in this study. The age of the patients varied from 12 to 64 years old, and seven of the patients were males (Table 1). The patients were treated for a malunion in various bones including the clavicle, humerus, radius, femur and tibia. Indication for corrective osteotomy was loss of function by decreased range of motion post-trauma due to acquired pathoanatomy in all patients with an upper limb deformity (*n* = 6). In the patients with a deformity of the lower limb (*n* = 4), the correction was performed due to complaints of pain and instability of the knee or ankle.

### 3.2. Accuracy

All corrective osteotomies were performed according to the predetermined plan without any complications. The preoperative deformities ranged from 7.1 to 27.5° in terms of angulation and 5.3 to 26.1° in terms of rotation. The achieved alignment deviated on average 2.1 ± 1.0 and 3.4 ± 1.6 degrees from the planning for the angulation and rotation, respectively (Table 2). In four cases, the achieved angulation was more than what was needed in the planned direction (overcorrection), and in another four cases, the angulation was less than what was needed (undercorrection). In terms of rotation, in four patients, the applied correction was more than intended, whereas in two patients, the correction was less. The achieved positioning of the bone deviated on average 1.8 mm from the intended position. Differences between planned and obtained angulation (*p* < 0.001), rotation (*p* = 0.009) and translation (*p* < 0.001) were found to be statistically significant.

### 3.3. Clinical Outcome

In all six patients who underwent a corrective osteotomy for a malunion of the upper limb, the range of motion was significantly improved after six months (Table 3). After correction of the malunions of the lower limb, all patients reported a significant reduction in pain and instability.

## 4. Discussion

Bone deformities in both the upper and lower extremities frequently lead to functional impairment, pain, instability and/or aesthetic concerns. Additionally, in the long term, this could lead to early-onset osteoarthritis of adjacent joints. Corrective osteotomy surgery of these post-traumatic acquired deformities can be very challenging since the deformation of the bone is often present in three dimensions. With the use of 3D-printing technologies, corrective osteotomy surgery has become more predictable. In this study, we presented our clinically applied two-step approach (cutting guide followed by reposition guide) of patient-specific 3D-planned corrective limb osteotomies. The results of this study show that our clinically applied method is reliable, feasible, user-friendly and accurate.

Several studies describe 3D-planned corrective limb osteotomies and show their technique to be feasible, leading to good functional outcomes [6,7,8,9,10,11,12,13,14]. Yet, even though the surgery is planned using state-of-the-art 3D software, postoperative evaluation is usually still performed in 2D on plain radiographs since postoperative CTs are not routinely made. Therefore, the majority of these studies evaluated their achieved accuracy based on postoperative radiographs, thereby only providing the accuracy in two dimensions: the anteroposterior and lateral direction [6,7,8,9,11]. Since the performed correction was planned in three dimensions, it is essential to assess the postoperative result in three dimensions as well to provide the accuracy of the performed correction and to gain insight into the cause of deviation in relation to suboptimal clinical outcomes. Omori et al. performed corrective osteotomies in 17 patients with a deformity of the humerus and compared postoperative 3D bone models with the preoperative planning using a surface registration technique [13]. In terms of translation, they showed a mean error of 1.7 mm in anterior–posterior translation, 1.3 mm in lateral–medial translation and 7.1 mm in proximal–distal translation, whereas they showed mean errors of 0.6°, 0.8° and 2.9° for varus–valgus, flexion–extension and internal–external rotation, respectively. Additionally, Dobbe et al. successfully performed a 3D-guided corrective osteotomy using a patient-specific plate in seven patients with post-traumatic distal radius deformities and showed a median residual translation and rotation error of 3.0 mm and 8.5°, respectively [12]. In addition to these scarce studies, this current study is one of the few studies which assessed the postoperative result in 3D. Where the previous studies were limited to one specific malunited bone, in this study, we performed 3D-planned corrective limb osteotomies in different body regions of ten patients, indicating the wide applicability of our technique. The results of this study show similar accuracy compared to previous reports with an average angulation of 2.1 ± 1.0°, rotation of 3.4 ± 1.6° and translation of 1.8 ± 1.8 mm.

The rationale behind using patient-specific 3D-printed guides is that it helps the surgeon perform the osteotomy and predrill the screws more reliably and according to the plan, leading to a more accurate correction. However, one of the possible pitfalls is translation of bone fragments over the osteotomy planes due to applied uncontrolled compression on the plate [10]. This is especially true in cases with extensive deformation of the bone where there is no good fit between the plate and bone. A suboptimal correction in these patients may result in residual functional impairment, pain and joint instability. Our two-step approach provides a solution for these patients by not predrilling the screws but using K-wires to secure the cutting guide, and subsequentially a reposition guide with these positioned K-wires, which forces bone fragments into the correct 3D-planned alignment while serving as a temporary fixation as well. The plate can generally be placed under the reposition guide for easy application. The specific design of the reposition guide allows for at least two screws to be drilled and placed both distal and proximal to the osteotomy level, which then hold the reposition while the guide is removed. Definitive fixation can then be performed by placing the remaining screws within the plate.

One of the limitations of this pilot study is the relatively small patient group. Since the goal of this study was to critically evaluate the accuracy of our two-step approach, we included all patients who were treated with this technique irrespective of anatomical site or nature of the post-traumatic deformity. Even though this study showed that our technique is clinically feasible and accurate in both upper and lower extremity deformities, the inclusion of different osteotomy locations led to a highly heterogeneous study population. Yet, one could also argue this a study strength, as it improves the external validity of our results regardless of the anatomical site. Additionally, since the primary aim of this research was to assess the accuracy of this technique, the clinical outcome in this study was limited to the range of motion, which was only of importance in the cases with upper limb deformities. Further studies should also incorporate patient-reported outcomes to also fully assess the impact of this method on functional recovery. This is especially true in patients with lower limb deformities, since these patients were not affected by a restricted range of motion.

Even though high accuracy of the planned correction was achieved in all patients, some minimal residual angulation, rotation and translation error were still present, although we argue this is clinically not relevant to patients’ functional outcome. At this stage, it is impossible to assess at what part of the surgery the error happened (e.g., positioning of the guides, securing the screws). In order to further improve the current method, it would be recommended to investigate each specific step within the procedure. In particular, the positioning of the cutting guide along the longitudinal axis is usually quite challenging in the case anatomical landmarks for verifying the correct position of the guides are limited (e.g., shaft fractures). Where the proximal or distal end of the bone generally has quite distinguishable features, the fit of the guide on the midshaft bone is usually less rigid. Further investigation on what impact guide positioning has on the accuracy is therefore recommended. In addition, this study included patients with relatively severe deformities with an average angulation of 15.5 ± 5.7° and rotation of 21.0 ± 7.9°. Patients with more subtile deformities might also benefit from 3D-guided corrective osteotomy surgery. To our knowledge, no clear cut-off for the point at which a deformity is too small to correct accurately has been established. Therefore, it further investigation of what deformities can accurately be corrected in order to utilize this technique to its full potential is also recommended.

## 5. Conclusions

This study showed that a two-step approach for 3D-guided patient-specific corrective limb osteotomies is reliable, feasible, user-friendly and accurate for corrective osteotomies of deformities of all long bones.

## Figures and Tables

**Figure 1 jpm-12-01458-f001:**
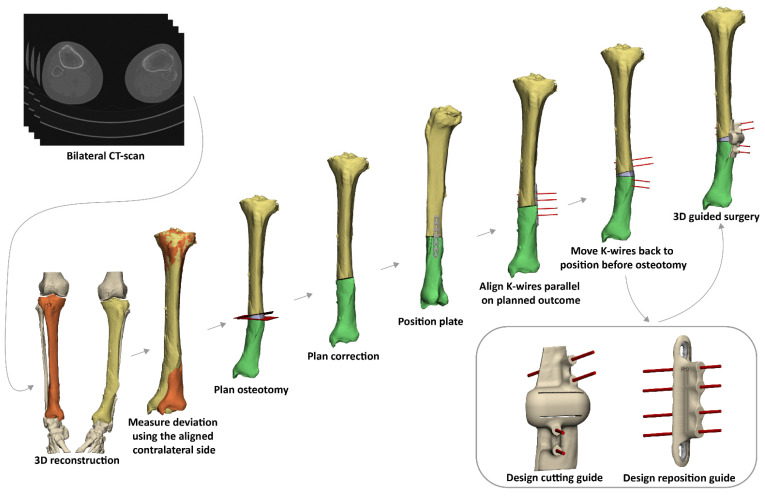
Workflow of a 3D-guided patient-specific corrective osteotomy using a two-step approach. (1) First, a 3D reconstruction is made from a bilateral CT scan. (2) By mirroring and aligning the contralateral (healthy) side on the malunited bone (orange), the deviation is measured. (3) Based on the deviation, the osteotomy and the correction are planned. (4) An osteosynthesis plate is chosen and positioned on the bone after correction. (5) K-wires are positioned parallel on the planned correction. (6) K-wires are placed parallel on the corrected bone, duplicated and moved to the corresponding position on the malunited bone before the correction is performed. (7) Patient-specific cutting and reposition guides are designed. (8) 3D-guided osteotomy is performed using the patient-specific cutting guide. Subsequently, the cutting guide is removed and the reposition guide (including the plate) is slid over the K-wires to achieve the intended correction.

**Figure 2 jpm-12-01458-f002:**
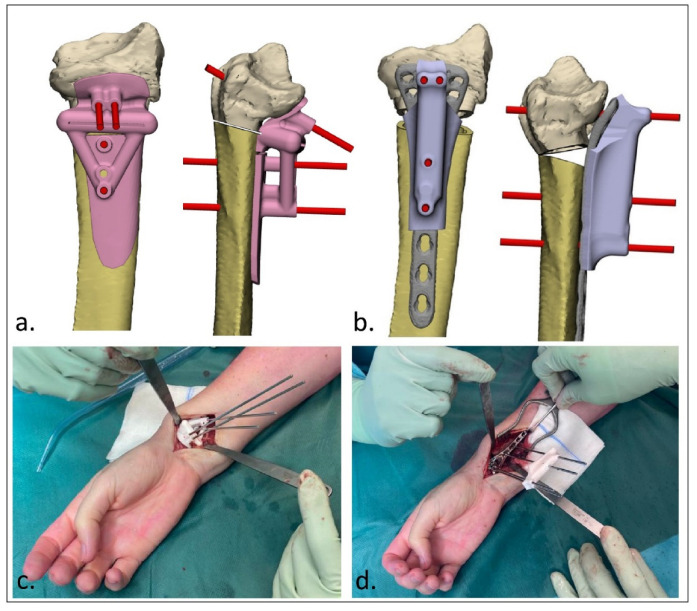
3D-guided patient-specific corrective osteotomy of a malunited distal radius that was initially treated conservatively in a cast (Case 6). (**a**) Frontal and lateral view of the designed cutting guide (pink) with the -wires (red); (**b**) frontal and lateral view of the designed reposition guide (purple) with the parallel K-wires (red); (**c**) operative usage of the cutting guide; (**d**) operative usage of the reposition guide. The specific design of the reposition guide allowed for at least two screws to be drilled and placed both distal and proximal to the osteotomy level. Note the convergent K-wires in the cutting guide (**c**), and the parallel K-wires as reduction aids in the reposition guide (**d**).

**Figure 3 jpm-12-01458-f003:**
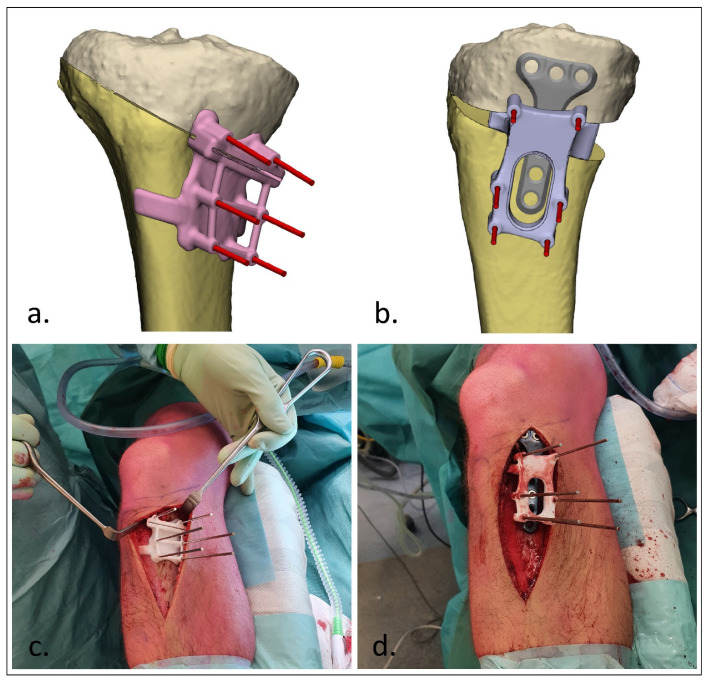
3D-guided patient-specific corrective osteotomy of a proximal tibia (Case 8). (**a**) Frontal view of the designed cutting guide (pink) with the K-wires; (**b**) lateral view of the designed reposition guide (purple) with the parallel K-wires and insertion of the planned wedge; (**c**) operative usage of the cutting guide; (**d**) operative usage of the reposition guide.

**Figure 4 jpm-12-01458-f004:**
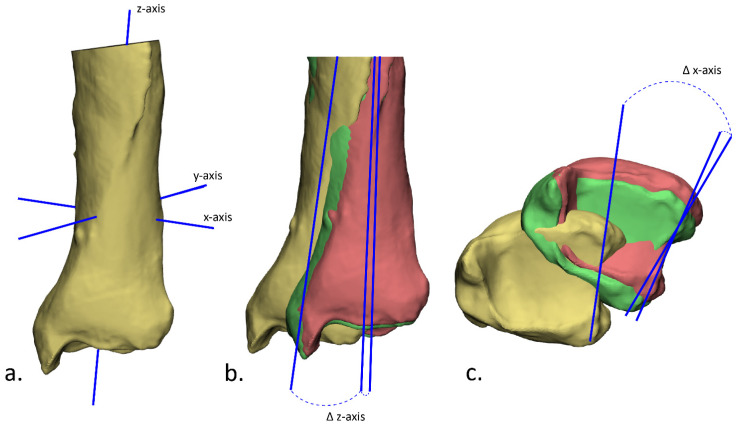
Evaluation of the achieved correction in terms of angulation and rotation. (**a**) First, the inertia axes were determined; (**b**) the angulation was then determined by measuring the angle between the z-axis of the preoperative (yellow) and planned (green) position of the bone, and between the planned and the postoperative (red) position; (**c**) the rotation was determined by measuring the angle between the x-axis of the preoperative and planned position, and the planned and postoperative position.

**Table 1 jpm-12-01458-t001:** Patient characteristics.

	Case	Sex	Age at Time of Surgery (Years)	Area of Deformity	Deformity	Planned Correction
**Upper limb**	1	F	16	Clavicula	Angulation; Shortening	Closed wedge
	2	F	23	Proximal humerus	Varus; Rotation	Closed wedge
	3	M	12	Midshaft radius	Angulation	Closed wedge
	4	M	16	Distal radius	Rotation	Rotation
	5	M	17	Distal radius	Volar angulation	Open wedge
	6	F	59	Distal radius	Volar angulation	Open wedge
**Lower limb**	7	M	28	Femur	Rotation	Rotation
	8	M	17	Proximal tibia	Varus; Increased tibial slope	Open wedge
	9	M	24	Proximal tibia	Increased tibial slope	Closed wedge
	10	M	64	Distal tibia	Varus; Rotation	Closed wedge

**Table 2 jpm-12-01458-t002:** Postoperative evaluation of the performed corrective osteotomies. Accuracy was assessed in terms of angulation (degrees), rotation (degrees) and translation (millimeters).

Case	Angulation (°)			Rotation (°)			Translation (mm)	
	Preoperative vs. Plan	Postoperative vs. Plan	Over/under Correction	Preoperative vs. Plan	Postoperative vs. Plan	Over/under Correction	Preoperative vs. Plan	Postoperative vs. Plan
1	20.5	3.7	Under	-	-	-	8.1	0.3
2	14.7	2.7	Under	22.8	1.3	Over	6.2	3.2
3	14.7	1.3	Over	-	-	-	7.2	0.8
4	-	-	-	26.1	4.3	Over	4.4	0.5
5	12.5	2.7	Over	-	-	-	7.5	1.6
6	27.5	1.8	Over	5.3	1.6	Over	7.1	1.0
7	-	-	-	26.0	5.1	Under	1.1	0.8
8	13.5	2.6	Under	-	-	Over	37.8	6.2
9	7.1	0.2	Under	-	-	-	3.5	0.4
10	13.6	1.6	Over	24.7	4.8	Under	17.2	2.9
**Average**	15.5 ± 5.7	2.1 ± 1.0		21.0 ± 7.9	3.4 ± 1.6		10.0 ± 10.1	1.8 ± 1.8

**Table 3 jpm-12-01458-t003:** Range of motion in the upper extremities before and 6 months after correction.

	Case	Restricted Joint	Preoperative Range of Restricted Motion	Postoperative Range of Motion (6 Months)
**Upper limb**	1	Clavicula	F/E 110-0-40; Ab/Ad 140-0-30	F/E 160-0-40; Ab/Ad 160-0-30
	2	Proximal humerus	F/E 150-0-40; Ab/Ad: 140-0-30; ER/IR: 90-0-L5	F/E 170-0-40; Ab/Ad: 160-0-30; ER/IR: 80-0-T12
	3	Midshaft radius	P/S: 10-0-60	P/S: 45/0/70
	4	Distal radius	P/S: 70-0-55	P/S: 70-0-80
	5	Distal radius	P/S: 70-0-30	P/S: 70-0-80
	6	Distal radius	F/E: 50/0/20 P/S: 20/0/80	F/E: 40/0/50 P/S: 60/0/80

F/E: flexion/extension, Ab/Ad: abduction/adduction, ER/IR: external rotation/internal rotation, P/S: pronation/supination, L5: 5th lumbar vertebra, T12: 12th thoracic vertebra.

## Data Availability

The authors declare that the data supporting the findings of this study are available within the paper.

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
