# Peer review of "A Two-Step Approach for 3D-Guided Patient-Specific Corrective Limb Osteotomies"

_jpm, 2022, doi:10.3390/jpm12091458_

Round 1

Reviewer 1 Report

The present paper is well written containing important informations for the interested surgeons. The method can be promoted and used by other doctors. 

I hope you will continue the study and provide us the outcome .

Reviewer 2 Report

The reference list of the manuscript contains only 13 titles, and is without inappropriate self-citations.I would encourage the authors to add to the literature especially on the popular patient specific method.

The manuscript is clear, with a good rate of novelty and significance. The manuscript present scientific resound and the design appropriate to test the hypothesis. The methods and software are clear described, with sufficient details to permit another researcher to reproduce the results. All aspects regarding the figures/tables/images are appropriate, and they are easy to interpret and understand. The presentation and the analyzed date are written in proper way. The presentation of the results are at high standard, with appropriate statistics. The results offer a development in the present knowledge, are significant, and are suitable interpreted.

Some notes are made as comments in the uploaded pdf.

Reviewer 3 Report

Dear Authors:

       I want to congratulate with You for Your work (clinical, and academic). In my opinion, some corrections are needed:

- table 3: external rotation / internal rotation better than exo/endo (ER/IR);

- line 172 maleS;

- line 175 "space" between last letter and "(";

- 2D and 3D first time should be written extensively; same for K-wire: I would write "Kirschner-wire (K-wire)" better than "K(isrchner)-wire";

- could You choose plates to be used, or there was a joint venture between a specific plate company and the planning&guiding company?

- there were a statistically significant difference between the planned and the obtained post-op axes?

- lenghtening/shortening was not consider, but both opening and closing wedge osteotomies would imply length discrepancy: could You recover length values as well?

- all presented cases were treated with a single plane osteotomy: do You use different  correction methods (hexapods ExFix, monoaxyal dynamic ExFix?) for more complex corrections? can You specify it into introduction or methods (10 cases out of XX osteotomies performed..)

best regards,
